# α-Lipoic acid prevents against cisplatin cytotoxicity via activation of the NRF2/HO-1 antioxidant pathway

**Joohyung Lee**, **So-Young Jung, Keum-Jin Yang, Yoonho Kim, Dohee Lee, Min Hyeong Lee, Dong-Kee Kim** *

Department of Otolaryngology, Daejeon St. Mary's Hospital, College of Medicine, The Catholic University of Korea, Daejeon, Republic of Korea

* cider12@catholic.ac.kr

**Data Availability Statement:** All relevant data are within the manuscript and its Supporting Information files.

## Abstract

The production of reactive oxygen species (ROS) by cisplatin is one of the major mechanisms of cisplatin-induced cytotoxicity. We examined the preventive effect of α-lipoic acid (LA) on cisplatin-induced toxicity via its antioxidant effects on *in vitro* and *ex vivo* culture systems. To elucidate the mechanism of the antioxidant activity of LA, NRF2 was inhibited using NRF2 siRNA, and the change in antioxidant activity of LA was characterized. MTT assays showed that LA was safe at concentrations up to 0.5 mM in HEI-OC1 cells and had a protective effect against cisplatin-induced cytotoxicity. Intracellular ROS production in HEI-OC1 cells was rapidly increased by cisplatin for up to 48 h. However, treatment with LA significantly reduced the production of ROS and increased the expression of the antioxidant proteins HO-1 and SOD1. E*x vivo*, the organs of Corti of the group pretreated with LA exhibited better preservation than the group that received cisplatin alone. We also confirmed the nuclear translocation of NRF2 after LA administration, and that NRF2 inhibition decreased the antioxidant activity of LA. Together, these results indicate that the antioxidant activity of LA was through the activation of the NRF2/HO-1 antioxidant pathway.

## Introduction

Cisplatin (cis-diamminedichloroplatinum II) is widely used as a chemotherapeutic agent to effectively treat various cancers. However, cisplatin therapy is limited by cellular resistance and severe side effects in normal tissues, including nephrotoxicity, neurotoxicity, and cytotoxicity [1, 2]. More than 60% of cisplatin-treated pediatric patients acquired bilateral toxicity in the conventional treatment range during chemotherapy [3]. Even when they grew up, severe hearing loss, in which the patients require a hearing aid or go deaf, was detected in 36% of CNS tumor survivors and 39% of non-CNS tumor survivors. Serious hearing loss in these patients is associated with a reduction in their social attainment [4]. If hearing loss could be prevented in the patients, many of the socioeconomic costs caused by hearing loss could be saved.

**Funding:** This research was supported by the Basic Science Research Program through the National Research Foundation of Korea (NRF) and funded by the Ministry of Education (NRF-2017R1D1A1B03027894 for D.K.K) and the Catholic Medical Center Research Foundation in program year 2017. (for D.K.K). The funders had no role in study design, data collection and analysis, decision to publish, or preparation of the manuscript.

**Competing interests:** The authors have declared that no competing interests exist.

Many studies have reported that cisplatin-induced cytotoxicity predominantly involves sensory hair cell death within the cochlea, and cells at the basal turn of the cochlea exhibit greater degeneration than those at the apical turn [5, 6]. The production of reactive oxygen species (ROS) by cisplatin is considered the main mechanism of action [6, 7]. In cochlear cells, NADPH oxidase 3 (NOX3), an isoform of nicotinamide adenine dinucleotide phosphate (NADPH) oxidase, is a major source of ROS. NOX3 is highly expressed in the organ of Corti and spiral ganglion [8, 9]. Cisplatin activates NOX3, although the mode of activation remains unclear [1].

Many studies have investigated the reduction of cisplatin cytotoxicity using antioxidants. Glutathione [10, 11], vitamin E [12, 13], N-acetylcysteine [14], D-methionine [15], and sodium thiosulfate [16] have been used in basic research, preclinical studies, and clinical studies to show protective effects against cisplatin toxicity. In particular, a recent clinical study in 109 patients <18 years of age with hepatoblastomas reported that treatment with sodium thiosulfate significantly lowered the rate of cisplatin-induced hearing loss, but did not lower the success rate of cancer treatment [16]

Even though sodium thiosulfate has shown good results in one recent clinical study, ~~but~~ no other clinical studies have shown meaningful results in preventing ototoxicity by cisplatin. Therefore, search for the effective ototoxicity preventive drugs is still ongoing. Among them, Alpha-lipoic acid (LA) showed promising results in recent in vitro and in vivo studies. Especially, it showed superior ototoxicity prevention effect compared to glutathione, one of the representative antioxidants. [17–19]. However, there is still not much research on the prevention of ototoxicity in LA, and so the effects of LA needs to be validated again; in particular, the mechanism by which LA prevents ototoxicity is not well known.

The antioxidant activity of LA is known to be represented by free radical quenching, metal chelation, and antioxidant recycling, but the exact mechanism is still unknown. [20, 21]. Recently, the neuroprotective effects of LA in cerebral ischemia injury animal models have been reported to reduce oxidative damage caused by ischemic stroke through the Nrf2 / HO-1 antioxidant pathway [22]. Similar results on the mechanism of antioxidant activity in LA have also been reported in animal models of traumatic brain injury [23]. So in this study we verified the preventive effect of LA on cisplatin-induced cytotoxicity using in vitro and ex vivo culture systems, and investigated whether LA acts by the Nrf2 / HO-1 antioxidant pathway, to prevent cisplatin-induced ototoxicity.

## Materials and methods

### *In vitro* tests

**HEI-OC1 cell culture.** The immortalized mouse organ of Corti cell line, HEI-OC1, provided by Dr. MK Park (Seoul National University College of Medicine, Seoul, Republic of Korea) was used for *in vitro* tests. The establishment and characterization of the HEI-OC1 cell line have been described previously [24]. HEI-OC1 cells were grown and passaged in Dulbecco's Modified Eagle's Medium, supplemented with 10% fetal bovine serum, 50 U/mL recombinant mouse interferon-γ, and 10 ng/mL ampicillin, and then cultured in a humidified 10% $CO_2$ environment at 33°C.

**MTT assay.** First, the maximum safe concentration of LA (Sigma-Aldrich, St. Louis, MO, USA) in HEI-OC1 cells was determined using the MTT assay, and then the protective effect of LA against cisplatin (Sigma-Aldrich) at various concentrations below the maximum safe concentration was determined.

HEI-OC1 cells were plated in 96-well plates at $1 \times 10^4$ cells/well in 0.1 mL of complete growth medium and incubated to 80% confluency over 24 h. The cells were then incubated

with various concentrations of LA (0.1–2 mM) for 24 h. Cell viability was determined using a commercial MTT assay, in accordance with the manufacturer's protocol (EZ-Cytox; Daeil Lab, Seoul, Republic of Korea). After incubation, optical densities were determined at 450 nm using a microplate reader (Bio-Rad, Hercules, CA, USA).

To examine the protective effect of LA against cisplatin-induced cytotoxicity, the cells were pretreated with LA at different concentrations for 1 h and then exposed to 20 μM cisplatin for 24 h.

**Intracellular ROS measurements.** In HEI-OC1 cells were divided into two groups; one with pretreatment of 0.5mM LA for 1 hour, and the other without pretreatment of LA. 20 μM cisplatin was treated to the cells for 16, 24 and 48 hours. The intracellular ROS was observed by DCFH-DA assay. (Sigma-Aldrich) [17]. Subsequently, the cells were harvested, and fluorescence signals were visualized using H2DCFDA (Thermo Fisher Scientific, Waltham, CA) according to the manufacturer's protocol. The green fluorescence from the cells was confirmed by fluorescence microscopy (Eclipse TE300, Nikon, Japan) and by fluorescence-activated cell sorting (FACS Canto™, Becton Dickinson and Company, Franklin Lakes, NJ).

**Western blotting.** HEI-OC1 cells were cultured for 16 h and then treated with 20 μM cisplatin for 24 h with or without pretreatment with 0.5 mM LA. Cell lysates were prepared using RIPA buffer (20 mM Tris-HCl, pH 7.4, 0.01 mM EDTA, 150 mM NaCl, 1 mM phenylmethylsulfonyl fluoride, 1 μg/mL leupeptin, and 1 mM Na$_3$VO$_4$), then resolved by sodium dodecyl sulfate-polyacrylamide gel electrophoresis. Equal amounts of protein were electroblotted onto a nitrocellulose membrane and then incubated with primary antibodies against superoxide dismutase (SOD) 1 (Cell Signaling Technology, Danvers, MA, USA), SOD2 (Novus Biologicals, Littleton, CO, USA), NRF2 (Cell signaling technology, Danvers, MA, USA) and heme oxygenase 1 (HO-1) (Cell Signaling Technology). Positive bands were visualized and analyzed using the ChemiDoc XRS Image system (Bio-Rad).

**Quantitative real-time polymerase chain reaction (PCR).** To compare the levels of inflammatory cytokines after treatment with LA, total RNA was isolated from the HEI-OC1 cells using a NucleoSpin RNA II kit (Macherey-Nagel, Düren, Germany). The cDNA was synthesized using a Reverse Transcriptase Premix (Elpis Biotech, Daejeon, Republic of Korea) and amplified via polymerase chain reaction (PCR) using Power SYBR Green Master Mix (Applied Biosystems, Foster City, CA, USA) with gene-specific primer sets. Quantitative real-time PCR (RT-PCR) was performed using the ABI 7500 FAST instrument (Applied Biosystems). The relative levels of mRNA were normalized to levels of glyceraldehyde 3-phosphate dehydrogenase (GAPDH).

**NRF2 siRNA transfection.** To elucidate the mechanism of antioxidant activity of LA, nuclear factor erythroid 2-related factor 2 (NRF2) was inhibited using NRF2 siRNA in HEI-OC cells. The siRNA for the *Nrf2* gene (Invitrogen, Carlsbad, CA, USA) was transfected into HEI-OC1 cells with Lipofectamine RNAiMax (Invitrogen, Carlsbad, CA) for 24 h, in accordance with the manufacturer's protocol. After NRF2 inhibition, changes in intracellular ROS, the expression levels of antioxidant proteins, and the levels of inflammatory cytokines were determined as described above.

**Extraction of nuclear and cytosol fractions.** To confirm the nuclear localization of the *Nrf2* gene, nuclear extraction from the cytoplasmic fraction of HEI-OC1 cells was performed using RSB solution [20 mM Tris-HCl (pH 7.4), 10 mM NaCl, and 2 mM MgCl$_2$]. After collecting cells by centrifugation at $600 \times g$, 10 volumes of RSB solution was then added for 5 min, followed by incubation for 12 min on ice. After homogenization 12 times using a dounce homogenizer, a clear extract was obtained by centrifugation at $300 \times g$ for 3 min. Western blotting was conducted as described above, and nuclear and cytosolic fractions were confirmed by histone H3 and GAPDH analyses.

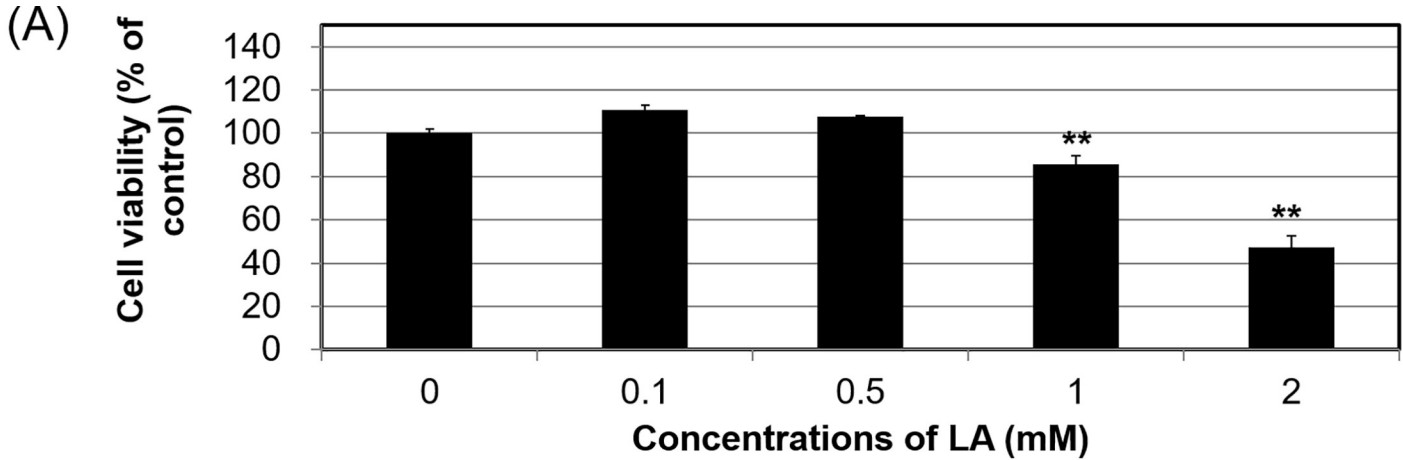

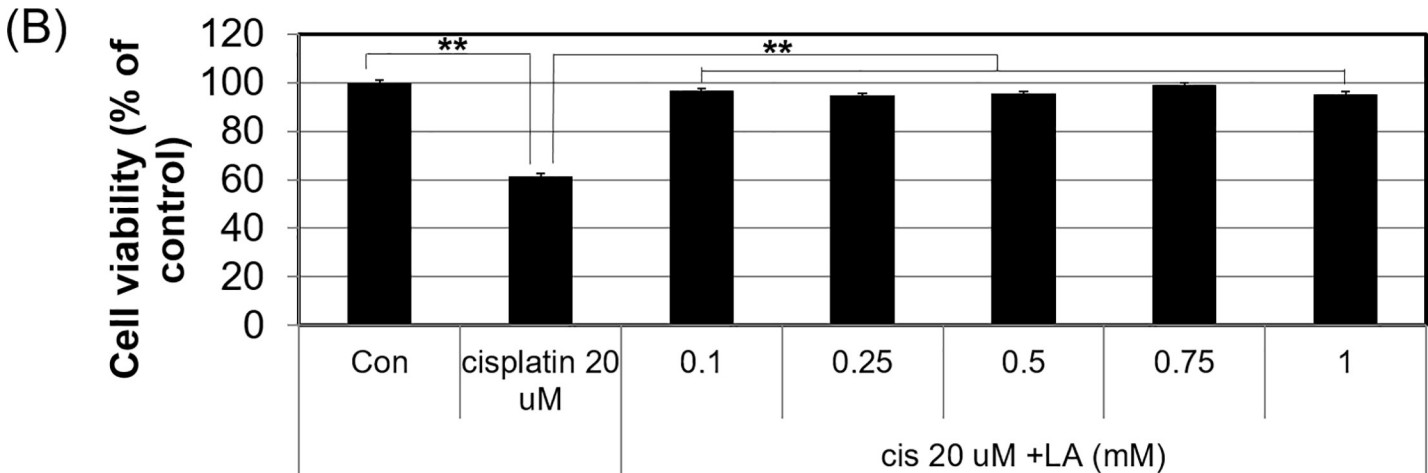

**Fig 1. Safety test of α-lipoic acid (LA) in HEI-OC1 cells and protective effect of LA against cisplatin-induced cytotoxicity.** HEI-OC1 cells were treated with various LA concentrations (0.1–2 mM) for 24 h. LA concentrations >1 mM induced significantly greater cytotoxicity compared with concentrations <0.5 mM (A). HEI-OC1 cells were pretreated with various concentrations of LA (0.1–1 mM) for 1 h, followed by 20 μM cisplatin for 24 h. LA provided significant protection against cytotoxicity induced by 20 μM cisplatin in the 0.1–1 mM treatment groups (B). **$P < 0.01$ compared with cisplatin alone as determined by an independent *t*-test.

### *Ex vivo* cochlear tests

All procedures involving animals were performed under the Laboratory Animals Welfare Act, the Guide for the Care and Use of Laboratory Animals, and the Guidelines and Policies for Rodent Experimentation provided by the Institutional Animal Care and Use Committee (IACUC) of the School of Medicine of the Catholic University of Korea. The study protocol was approved by the IACUC in School of Medicine, The Catholic University of Korea. (approval no. CMCDJ-AP-2018-001).

Animals were housed in groups of five and maintained in a temperature-controlled environment at 22˚C on a 12:12-h light:dark cycle. Primary cochlear explants were prepared from C57/BL6 mice (Damul science, Daejeon, Korea) on postnatal day 3 (P3). Cochlear explants from both cochlea of nine mouse pups were divided into three groups and used for the experiment. Mouse pups were euthanized by decapitation and the cochleae were dissected out. The dissected organs of Corti were incubated in high glucose DMEM containing 5% (v/v) fetal bovine serum, 5% (v/v) horse serum, and 10 ng/mL ampicillin at 37˚C in a 5% (v/v) $CO_2$ humidified incubator. They were then treated with 20 μM cisplatin for 24 h with or without

pretreatment with 0.5 mM LA for 1 h. After fixation in 4% (v/v) paraformaldehyde solution, permeabilization with acetone, antibody binding using Alexa Fluor 488 phalloidin (Molecular Probes, Eugene, OR, USA), and mounting in Vectashield Mounting Medium containing 4′,6-diamidino-2-phenylindole (DAPI) (Vector Laboratories, Burlingame, CA, USA), nano-particle uptake was measured by fluorescence microscopy (Eclipse TE300, low magnification; Nikon, Tokyo, Japan) and confocal microscopy (LSM880 with Airyscan; Carl Zeiss, Oberko-chen, Germany). All specimens were stained with DAPI and fluorescein isothiocyanate-labeled phalloidin to evaluate the status of hair cell cilia and tissue architecture.

### Statistical analysis

Statistical analysis was performed using the independent $t$-test and one-way analysis of variance, and each experiment was performed independently and repeated at least three times. A value of $P < 0.05$ was considered significant.

## Results

### Verification of the protective effects of LA against cisplatin-induced cytotoxicity

**Cytotoxicity of LA in HEI-OC1 cells and the protective effects against cisplatin-induced cytotoxicity.** LA did not show any toxicity in HEI-OC1 cells at concentrations up to 0.5 mM

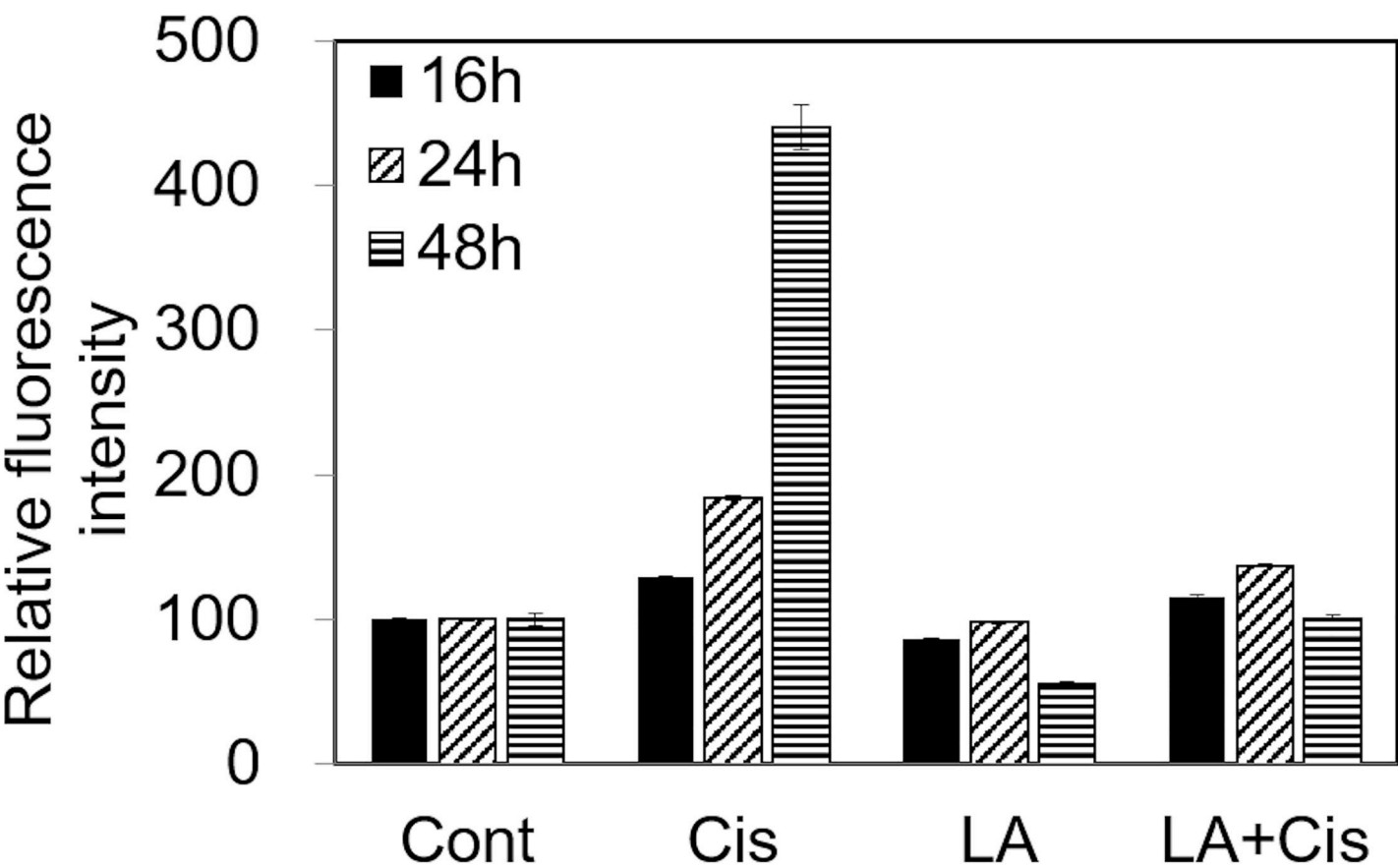

**Fig 2. The effect of α-lipoic acid (LA) on reactive oxygen species (ROS) production in auditory cells.** Intracellular ROS measured using FACScan flow cytometry showed that ROS production in HEI-OC1 cells increased by cisplatin up to 48 hours, but not in the LA-treated cells.

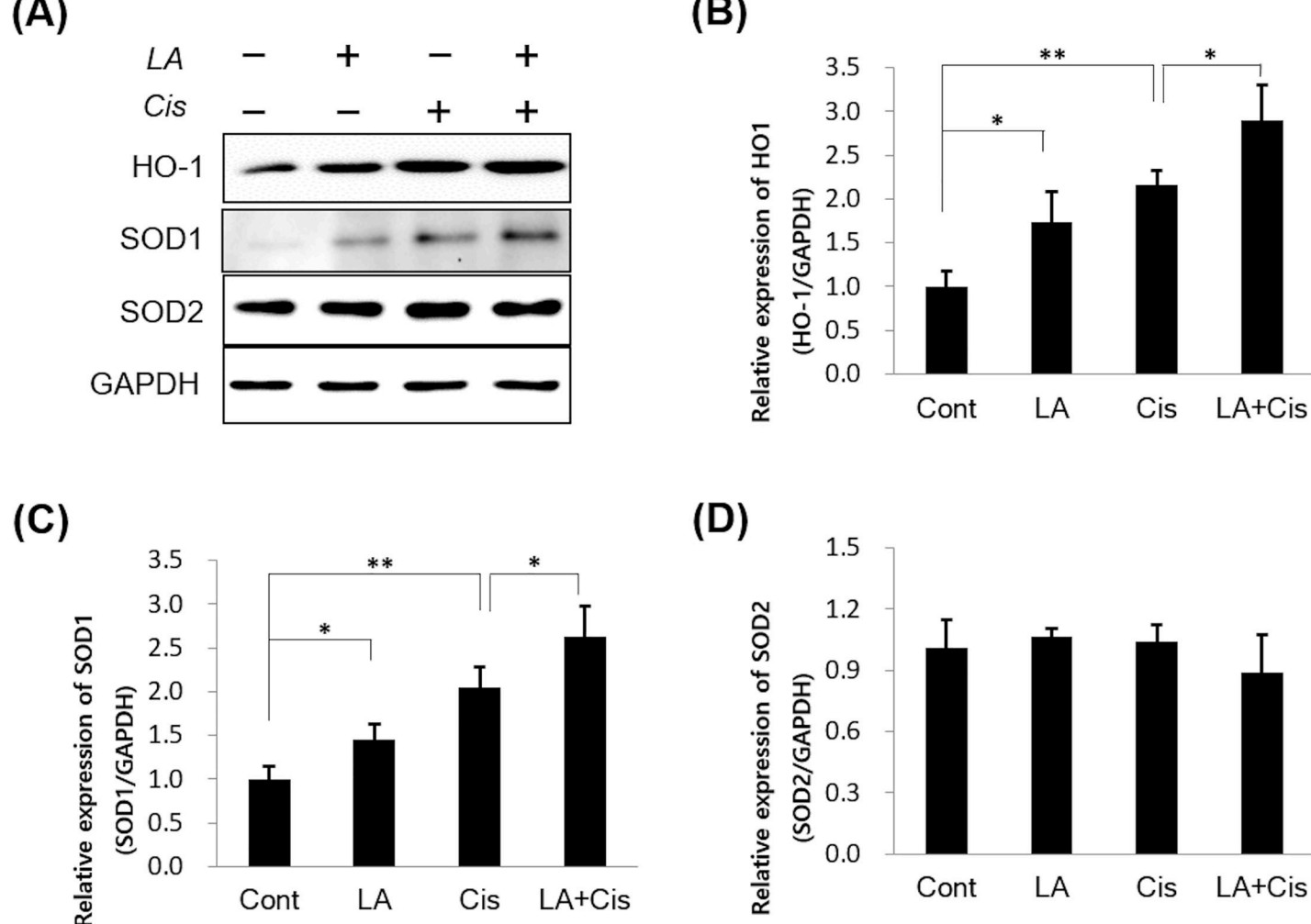

**Fig 3. The effects of α-lipoic acid (LA) on intracellular antioxidant proteins in cisplatin-treated HEI-OC1 cells.** HEI-OC1 cells were treated for 24 h with culture medium, dimethyl sulfoxide, 20 μM cisplatin, or 20 μM cisplatin with 0.5 mM LA pretreatment. The levels of intracellular antioxidant proteins were measured by western blot analysis. (A) Protein levels of the groups were compared and are represented by a bar graph (B-D). After treatment with cisplatin, HO-1 and SOD-1 expressions in HEI-OC1 cells were significantly increased compared to those in the control group; however, the expressions were further increased by pretreatment of LA. The data represent the mean ± SD of three independent experiments. $^*P < 0.05$; $^{**}P < 0.01$ compared with cisplatin alone as determined by an independent *t*-test.

(Fig 1A), and it showed a protective effect against cisplatin-induced cytotoxicity at concentrations up to 1.0 mM, whereas the cell viability did not decrease (Fig 1B). Based on these results, 0.5 mM LA concentration was considered safe in HEI-OC1 cells, and was also assumed to have a protective effect against cisplatin.

**Intracellular ROS measurements.** As shown in Fig 2, intracellular ROS increased continuously for 48 hours after 20μM cisplatin was treated in the cells. However, when LA was pretreated to cisplatin treatment, intracellular ROS did not increase and showed similar degree to control group.

**Western blotting.** To determine the cytoprotective effect of LA against cisplatin via its antioxidant activity, SOD1 and 2 and HO-1 levels were determined by western blotting. In the group pretreated with LA for 1 h, SOD1 and HO-1 levels were significantly higher than those in the control or cisplatin-alone groups (Fig 3). However, there were no significant differences in SOD2 levels among the three groups.

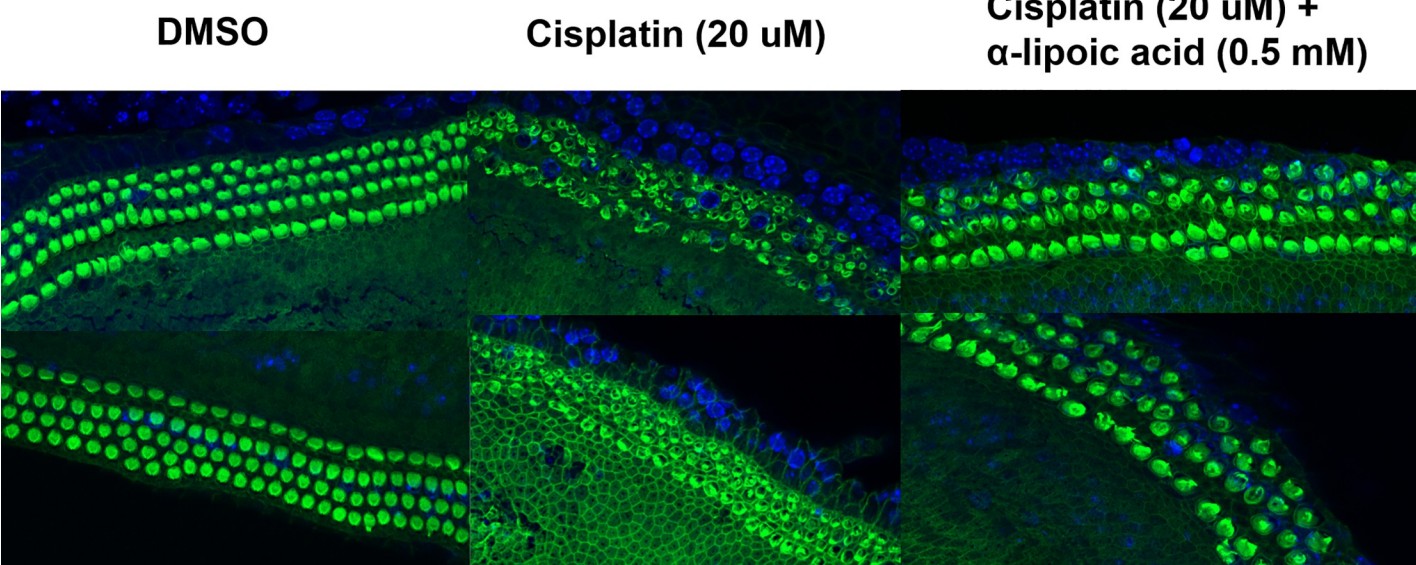

**Fig 4. Effect of α-lipoic acid (LA) on cisplatin-induced cytotoxicity in organ of Corti explants.** Organ of Corti explants were treated with culture medium, dimethyl sulfoxide, 20 μM cisplatin, or 20 μM cisplatin with 0.5 mM LA pretreatment. The explants were stained with Alexa Fluor 488-conjugated phalloidin and were then observed using confocal microscopy.

**Ex vivo cochlear tests.** To investigate the protective effects of LA against cisplatin in the cochlea, isolated organs of Corti were treated with 20 μM of cisplatin, and the status of hair cell cilia and tissue architecture at the middle turn were observed. The cochleae that were treated with 20 μM cisplatin alone exhibited disarrayed hair cells and significantly disrupted stereocilia bundles; however, the group co-treated with 0.5 mM LA exhibited an orderly arrangement of hair cells and remarkably preserved architecture of the organ of Corti (Fig 4).

## Identification of the antioxidant mechanism of LA

We observed that LA treatment induced a prominent increase in HO-1 expression using western blotting. The following experiments were conducted to confirm that this was due to the activation of the NRF2/HO-1 antioxidant pathway.

**Nuclear translocation of NRF2 by LA.** Normally NRF2 is negatively regulated by a cytosolic protein called KEAP1 (Kelch-like ECH associated protein 1). However, under the oxidative stressful conditions, NRF2 is released from KEAP1 and migrates from the cytoplasm to the nucleus to promote the expression of antioxidant proteins [25]. We observed whether the migration of NRF2 to the nucleus increased in HEI-OC1 cells, treated with LA. We also examined whether inhibition of NRF2 eliminated the antioxidant activity of LA.

HEI-OC1 cells treated or untreated with NRF2 siRNA were further treated with cisplatin and LA, and the expression of NRF2 was compared in the nuclei and cytoplasm. The relative nuclear to cytoplasmic NRF2 expression ratio was significantly increased by cisplatin, and was further increased by pretreatment with LA. However, this increase was reversed by NRF2 siRNA treatment (Fig 5A and 5B). The expression of HO-1 protein in HEI-OC1 cells showed similar results (Fig 5C and 5D). Together, these results show that LA acted through the activation of the NRF2/HO-1 pathway.

**Reduction of the antioxidant and anti-inflammatory actions of LA by NRF2 inhibition.** A change in the antioxidant activity of LA due to NRF2 inhibition was observed at 48 h, as an increase in cisplatin-induced intracellular ROS was prominent after 48 h. Similar to the

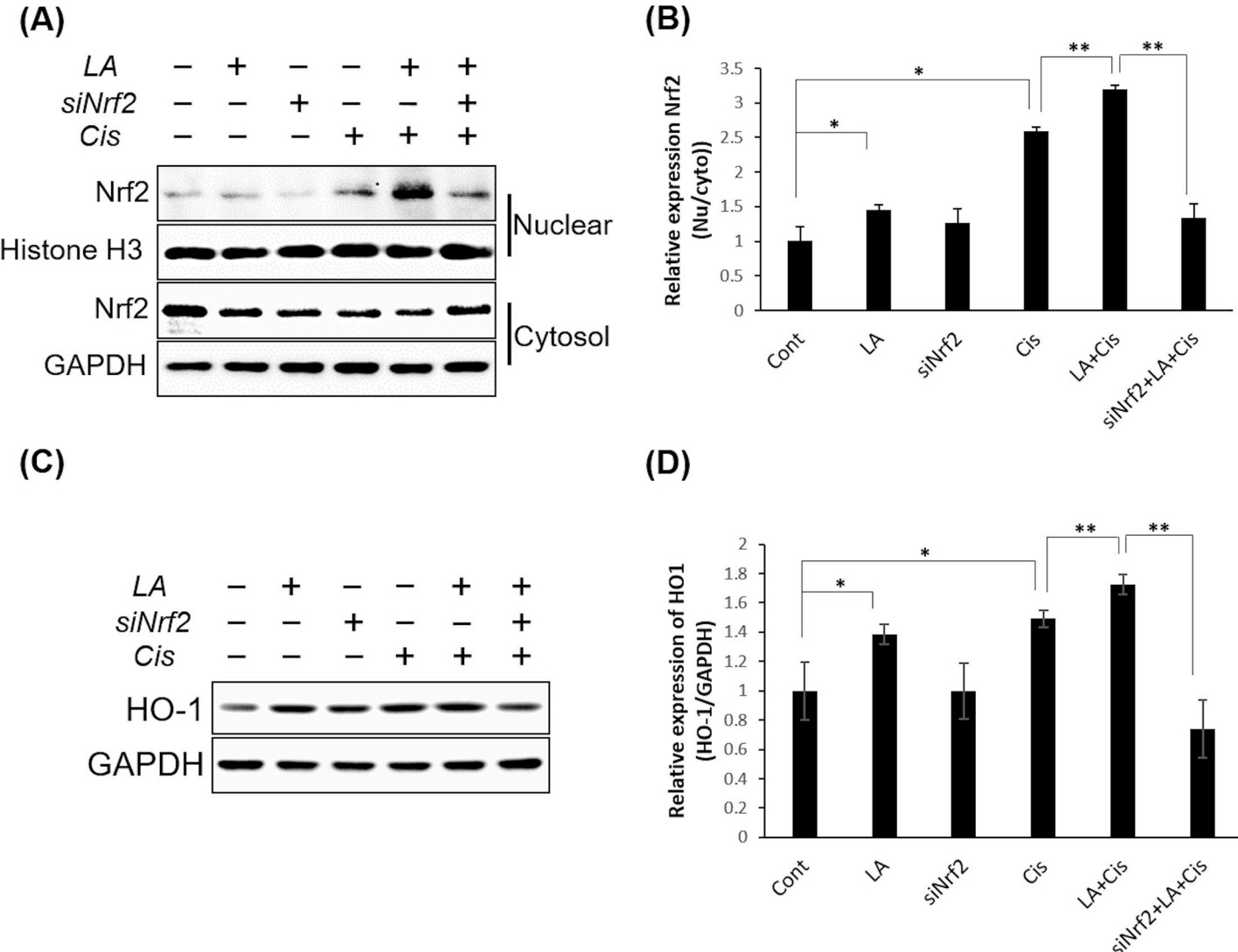

**Fig 5. Alpha-lipoic acid (LA) promoted the nuclear translocation of NRF2.** The relative nuclear to cytoplasmic NRF2 expression ratio was significantly increased by cisplatin and was further increased by pretreatment with 0.5 mM LA. However, this increase was reversed by siRNA treatment (A), (B). The expression of HO-1 protein in HEI-OC1 cells was also significantly increased by cisplatin and further increased by LA co-treatment, but not increased when cells were treated with siRNA (C), (D). Data are the means ± SD from three independent experiments performed in duplicate. $^*P < 0.05$; $^{**}P < 0.01$; $^{***}P < 0.001$.

previous results, the increase in ROS by cisplatin was reduced by LA pretreatment, but this effect of LA was offset by NRF2 inhibition (Fig 6). We then observed changes in the expression of inflammatory cytokines (Fig 7). After LA pretreatment, the mRNA levels of proinflammatory cytokines tumor necrosis factor-α (TNF-α) and interferon-γ were decreased, and the level of IL-10, an anti-inflammatory cytokine, was significantly increased in HEI-OC1 cells, but this effect of LA was offset by NRF2 inhibition.

## Discussion

Our experiments showed that LA prevented the toxic effects of cisplatin in the inner ear, both *in vitro* and *ex vivo*. In addition, the antioxidant effect of LA was shown to play a role in this prevention. In an *in vivo* study, administration of LA prevented cisplatin cytotoxicity and

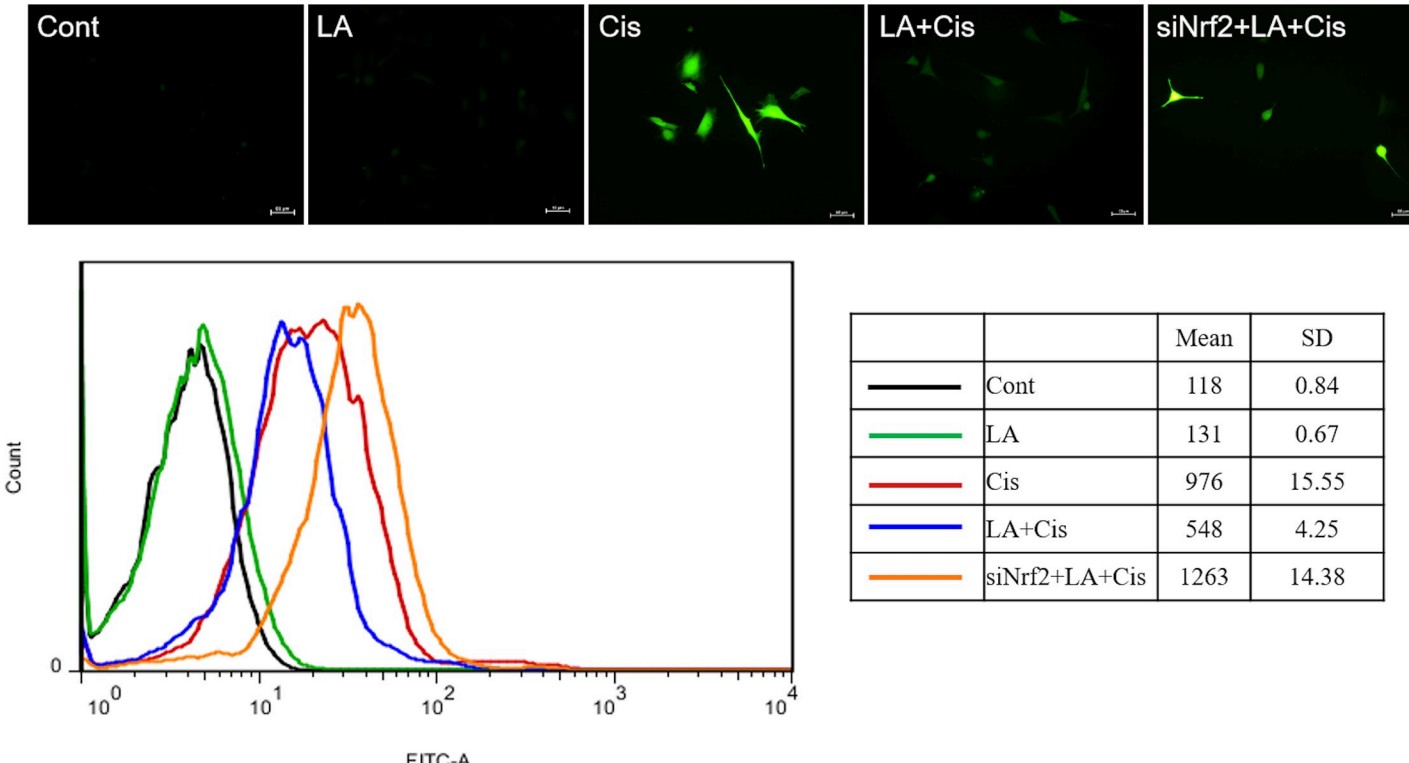

**Fig 6. Effect of NRF2 inhibition on the antioxidant effects of α-lipoic acid (LA) in auditory cells.** Reduction of intracellular ROS levels by LA pretreatment was reversed by NRF2 inhibition, demonstrating that LA acts through activation of the NRF2/HO-1 pathway.

preserved hearing [18, 26]. In two studies using distortion product otoacoustic emissions or the auditory brainstem response, hearing was preserved in animals with hearing loss treated with LA, but these studies did not fully explain the mechanism of antioxidant activity of LA. In the present study, we identified the antioxidant activity of LA in an inner ear cell line to understand its mechanism of action.

Our MTT results showed that LA was safe at doses up to 0.5 mM, whereas the toxic concentration of LA, when administered alone, was similar to or somewhat lower than that previously reported. A recent study reported prevention of hearing loss from cisplatin treatment via systemic administration of LA *in vivo*, and also showed that LA was not toxic in an inner ear cell line at concentrations up to 4 mM [18]. In the present study, pretreatment of HEI-OC1 cells with up to 1.0 mM LA before cisplatin administration prevented the reduction in cell viability due to cisplatin (Fig 1B). Fig 4 shows that LA had a protective effect against cisplatin in tissues. The group pretreated with LA exhibited remarkably preserved hair cells and architecture of the organ of Corti compared to those of the group that received cisplatin without pretreatment.

Figs 2 and 3 show the decrease in intracellular ROS and increases in HO-1 and SOD1 levels after LA pretreatment, with the expression of HO-1 more markedly increased. HO-1 is an enzyme that catalyzes the degradation of heme. Its expression is induced by oxidative stress in animal models, thus an increase in HO-1 expression confers protection [27]. In a recent study in an animal model of transient middle cerebral artery occlusion, LA treatment was neuroprotective and promoted functional recovery after ischemic stroke by attenuating oxidative damage, which was partially mediated by the NRF2/HO-1 pathway [22]. Our results showed that

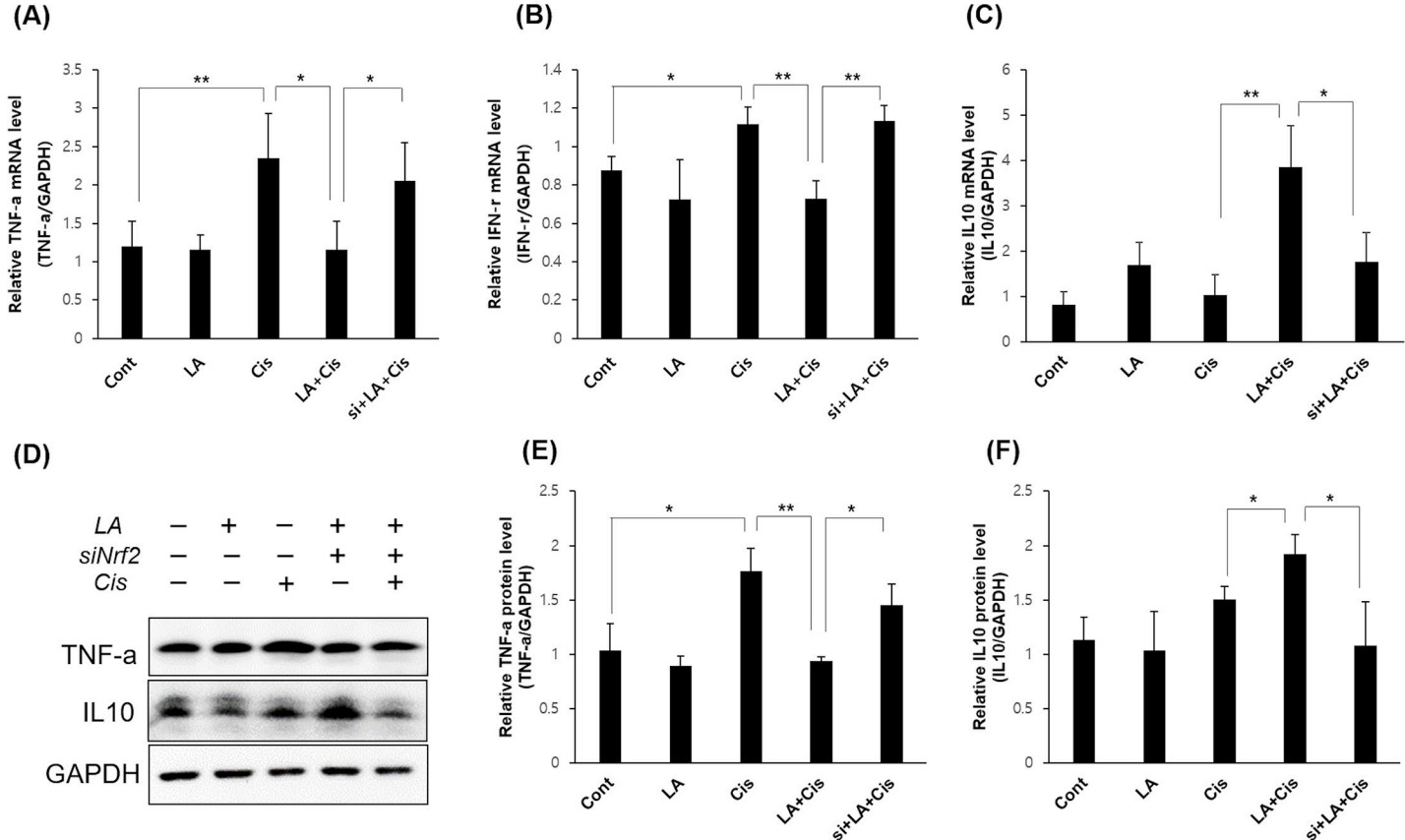

**Fig 7. The effects of α-lipoic acid (LA) on of inflammatory cytokines.** With 0.5 mM LA pretreatment, the mRNA levels of proinflammatory cytokines tumor necrosis factor-α (A) and interferon-γ (B) were decreased, and the level of IL-10 (C), an anti-inflammatory cytokine, was increased significantly in HEI-OC1 cells, but this action of LA was offset by NRF2 inhibition. The results of western blots of TNF alpha and IL10 were also similar to the results of PCR (D-F). The data represent the means ± SD of three independent experiments. $^{*}P < 0.05$; $^{**}P < 0.01$; $^{***}P < 0.001$.

activation of the NRF2/HO-1 pathway was also applicable to inner ear cells, which was confirmed by the observation that LA treatment induced nuclear translocation of NRF2. Furthermore, this process was inhibited by treatment with NRF2 siRNA (Fig 5). In addition, the antioxidant and anti-inflammatory effects of LA were also offset by NRF2 inhibition (Figs 6 and 7).

Activation of the NRF2/HO-1 pathway is also known to be associated with the expression of other antioxidant proteins, including SOD1 [28]. SOD1 binds copper and zinc ions and is one of three SODs responsible for destroying free superoxide radicals in the body. The encoded isozyme is a soluble cytoplasmic and mitochondrial intermembrane space protein, acting as a homodimer to convert naturally occurring but harmful superoxide radicals to molecular oxygen and hydrogen peroxide [29]. Similar to our results, several studies of various organs, including the brain and testis, showed that administration of LA increased SOD1 and protected against oxidative stress [30, 31].

In conclusion, LA effectively reduced cisplatin-induced ROS production and increased cell viability in an inner ear cell line. The activity of LA was mediated by activation of the NRF2/HO-1 pathway. Considering the safety of LA in the inner ear cell line and the protective effects of LA against cisplatin cytotoxicity, LA may be an effective treatment for the prevention of cytotoxic hearing loss.

## Supporting information

**S1 Fig. Uncropped and unadjusted images for Western blot of Fig 3A.**
(PDF)

**S2 Fig. Uncropped and unadjusted images for Western blot of Fig 5A.**
(PDF)

**S3 Fig. Uncropped and unadjusted images for Western blot of Fig 5C.**
(PDF)

**S4 Fig. Uncropped and unadjusted images for Western blot of Fig 7D.**
(PDF)

## Acknowledgments

This research was supported by the Basic Science Research Program through the National Research Foundation of Korea (NRF) and funded by the Ministry of Education (NRF-2017R1D1A1B03027894) and the Catholic Medical Center Research Foundation in program year 2017.

## Author Contributions

**Conceptualization:** Joohyung Lee, Dong-Kee Kim.

**Investigation:** So-Young Jung, Keum-Jin Yang, Yoonho Kim, Dohee Lee, Min Hyeong Lee.

**Supervision:** Dong-Kee Kim.

**Validation:** Keum-Jin Yang, Yoonho Kim.

**Visualization:** Dohee Lee.

**Writing – original draft:** Joohyung Lee, So-Young Jung.

**Writing – review & editing:** Min Hyeong Lee, Dong-Kee Kim.

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
