## [Decision Letter · Decision Letter 0]

4 Sep 2019

PONE-D-19-21584

α-Lipoic acid prevents against cisplatin cytotoxicity via activation of the NRF2/HO-1 antioxidant pathway

PLOS ONE

Dear Dr. Kim,

Thank you for submitting your manuscript to PLOS ONE. After careful consideration, we feel that it has merit but does not fully meet PLOS ONE’s publication criteria as it currently stands. Therefore, we invite you to submit a revised version of the manuscript that addresses the points raised during the review process.

Your manuscript was reviewed by two experts and both of them suggested additional data to support your hypothesis.

We would appreciate receiving your revised manuscript by Oct 19 2019 11:59PM. To enhance the reproducibility of your results, we recommend that if applicable you deposit your laboratory protocols in protocols.io, where a protocol can be assigned its own identifier (DOI) such that it can be cited independently in the future. For instructions see: http://journals.plos.org/plosone/s/submission-guidelines#loc-laboratory-protocols

We look forward to receiving your revised manuscript.

Kind regards,

Partha Mukhopadhyay, Ph.D.

Academic Editor

PLOS ONE

Journal Requirements:

1, We noticed minor instances of text overlap with the following previous publication(s), which need to be addressed:

doi: 10.1016/j.bbrc.2014.04.118.

The text that needs to be addressed involves the intracellular ROS measurement methods and results sections

In your revision please ensure you cite all your sources (including your own works), and quote or rephrase any duplicated text outside the methods section. Further consideration is dependent on these concerns being addressed."

2. At this time, we request that you please report additional details in your Methods section regarding animal care, as per our editorial guidelines:

(a) Please state whether the provided ethics committee contains animal welfare experts or whether an animal ethics or IACUC committee reviewed and approved the study. Please provide the full name of the committee that reviewed and approved the study

(b) Please state the source and number of mice used in the study

(c) Please provide details of animal welfare (e.g., shelter, food, water, environmental enrichment)

(d) Please describe any steps taken to minimize animal suffering and distress, such as by administering anaesthesia

(e) Please include the method of euthanasia

(f) Please provide additional details regarding the care of the mice if they were housed prior to euthanasia or if the dams were kept prior to birth, including housing, environment, food and water.

Thank you for your attention to these requests.

Reviewers' comments:

Reviewer's Responses to Questions

**Comments to the Author**

1. Is the manuscript technically sound, and do the data support the conclusions?

Reviewer #1: Yes

Reviewer #2: Yes

2. Has the statistical analysis been performed appropriately and rigorously? 

Reviewer #1: Yes

Reviewer #2: Yes

3. Have the authors made all data underlying the findings in their manuscript fully available?

Reviewer #1: Yes

Reviewer #2: Yes

4. Is the manuscript presented in an intelligible fashion and written in standard English?

Reviewer #1: Yes

Reviewer #2: Yes

5. Review Comments to the Author

Reviewer #1: In this article, the authors studied the protective effect of alpha-lipoic acid against cisplatin induced cytotoxicity in vitro and ex vivo in mouse auditory sensory cells. The authors demonstrated that alpha-lipoic acid significantly reduced intracellular ROS production in cisplatin treated cells. However, for completeness of the experiments, I have the following comments:

1) In all groups of treatments, the effects of alpha-lipoic acid alone need to be shown.

2) Figure 5 is generally illegible. The general claim of the author is that alpha-lipoic acid promotes the nuclear translocation of Nrf2, however the western blot data is unclear in the amount of cytoplasmic Nrf2.

3) The general changes of inflammatory cytokines measured by PCR methods are inconsistent with the author's claims in the text. Furthermore, the authors need to demonstrate changes in cytokine by direct measurement and not just on the level of mRNA.

4) There is strong discrepancy between Figure 3A and 5C in terms of how cisplatin alone changes the level of HO-1.

Minor points:

1) Figure 2B has insufficient labelling to identify the treatment groups.

2) Figure 6B, the MFI and the histograms do not match.

Reviewer #2: 1)Introduction: Explain the rationale for focusing on the cochlea better. What is the occurrence rate of pediatric hearing loss resulting from cisplatin treatment?

2) Introduction: The authors mention several antioxidants and sodium thiosulfate which has specifically been used to ameliorate the effects of cisplatin induced hearing loss in a clinical study. Why then is alpha-lipoic acid better?

3) Introduction: Please give more background about oxidative stress response mechanisms such as the NRF2/HO1 pathway. Why did the authors choose to focus on this pathway specifically?

4) Please check and remedy the titles and labeling throughout methods and materials section and the results section as per PLOS One guidelines. Also, figure legends should be a separate section and not included within the results section.

5) Results: Explain better the significance of testing NRF2 nuclear to cytoplasmic ratio.

6) Figure 2B: Please provide a key for the graph curves depicted.

7) All figures should be properly labeled.

8) Figures are poor quality. Please check guidelines for figure submission.

9) Result 2-1: Nuclear translocation of NRF2 by LA: What is the figure pertaining to this? If this is the immunofluorescence image on the penultimate page, this is not properly explained nor labeled. What is the graph with curves depicting? Where is it described and where is the key?

10) Results: Figure 5A and 5B: Please include a LA only control and a siNRF2 only control in both these experiments.

6. PLOS authors have the option to publish the peer review history of their article (what does this mean?). If published, this will include your full peer review and any attached files.

Reviewer #1: No

Reviewer #2: No

---

## [Author Response · Author response to Decision Letter 0]

13 Nov 2019

Reviewer #1: In this article, the authors studied the protective effect of alpha-lipoic acid against cisplatin induced cytotoxicity in vitro and ex vivo in mouse auditory sensory cells. The authors demonstrated that alpha-lipoic acid significantly reduced intracellular ROS production in cisplatin treated cells. However, for completeness of the experiments, I have the following comments:

 1) In all groups of treatments, the effects of alpha-lipoic acid alone need to be shown.

Thank you for your comments. As you pointed out, we retested and modified Figures 3, 5, 6, and 7 with the addition of the LA only group.

 2) Figure 5 is generally illegible. The general claim of the author is that alpha-lipoic acid promotes the nuclear translocation of Nrf2, however the western blot data is unclear in the amount of cytoplasmic Nrf2.

Previously used NRF2 antibody (abcam) resulted in multi band in western blot and was not good, so we replaced it with a reference in the literature. We repurchased antibodies to NRF2 (cell signaling technology), retested them, and modified the results. In the retest results, a clearer blot of cytoplasmic NRF2 was obtained.

 3) The general changes of inflammatory cytokines measured by PCR methods are inconsistent with the author's claims in the text. Furthermore, the authors need to demonstrate changes in cytokine by direct measurement and not just on the level of mRNA.

We fixed the wrong text description for IL 10. As shown in the results, the treatment of LA increased the mRNA levels of anti-inflammatory cytokine IL10, but it was described as ‘decreased’ in the previous text. (Last sentence of the 'Reduction of the antioxidant and anti-inflammatory actions of LA by NRF2 inhibition' section of the Results.)

As you mentioned, the western blots of TNF alpha and IL10 were also added, and the results of western blots showed the same trend as the PCR results. (Figure 7D~F) Thank you!

 4) There is strong discrepancy between Figure 3A and 5C in terms of how cisplatin alone changes the level of HO-1.

The experiment was repeated and cisplatin alone treatment in OC1 cells resulted in a significant increase in HO1 expression, so the figure 3 and 5 were modified.

Minor points:

1) Figure 2B has insufficient labelling to identify the treatment groups.

Figure 2B shows the result of the bar graph of figure 2A at 24 hours. But the contents of figure 2B overlaps with figure 2A and the difference between the experimental groups is not visually significant. So we removed it. 

2) Figure 6B, the MFI and the histograms do not match.

Sorry for the mistake. We corrected it. 

 Reviewer #2: 1)Introduction: Explain the rationale for focusing on the cochlea better. What is the occurrence rate of pediatric hearing loss resulting from cisplatin treatment?

Thank you for your comments. As you pointed out, we added the prevalence of hearing loss by cisplatin to the introduction and also added why it is important to prevent hearing loss.

Added sentences:

Even when they grew to adults, severe hearing loss (that requiring a hearing aid or deafness) was detected in 36% of CNS tumor survivors and 39% of non-CNS tumor survivors. Serious hearing loss in these patients is associated with a reduction in their social attainment (4). If hearing loss could be prevented in these patients, many of the socioeconomic costs caused by hearing loss could be saved.

 2) Introduction: The authors mention several antioxidants and sodium thiosulfate which has specifically been used to ameliorate the effects of cisplatin induced hearing loss in a clinical study. Why then is alpha-lipoic acid better?

Many antioxidants have been tried to prevent ototoxic hearing loss, but none of them have had a satisfactory effect. So, it is necessary to keep trying new antioxidants for ototoxic hearing loss. Recent papers have shown promising results for ototoxic hearing loss of alpha lipoic acid. This study was conducted to re-verify this and to understand the mechanism.

Added sentences:

Even though sodium thiosulfate has shown good results in one recent clinical study, but no other clinical studies have shown significant results in preventing ototoxicity by cisplatin. Therefore, the search for effective ototoxicity preventive drugs is still ongoing. Among them, Alpha-lipoic acid (LA) showed promising results in recent in vitro and in vivo studies. Especially, it showed superior ototoxicity prevention effect compared to glutathione, one of the representative antioxidants. (17-19). But there is still not much research on the prevention of ototoxicity in LA, so it needs to be validated again. In particular, the mechanism by which LA prevents ototoxicity is not well known.

 3) Introduction: Please give more background about oxidative stress response mechanisms such as the NRF2/HO1 pathway. Why did the authors choose to focus on this pathway specifically?

We reviewed the results of a recent study on the mechanism of alpha lipoic acid in brain injury animals. We added an explanation for this.

Added sentences:

The antioxidant activity of LA is known to be represented by free radical quenching, metal chelation, and antioxidant recycling, but the exact mechanism is still unknown. (20, 21). Recently, the neuroprotective effects of LA in cerebral ischemia injury animal models have been reported to reduce oxidative damage caused by ischemic stroke through the Nrf2 / HO-1 antioxidant pathway (22). Similar results on the mechanism of antioxidant activity in LA have also been reported in animal models of traumatic brain injury (23). So, in this study, we verified the preventive effect of LA on cisplatin-induced cytotoxicity using in vitro and ex vivo culture systems, and investigated whether LA acts by this Nrf2 / HO-1 antioxidant pathway, as in brain injury animals, to prevent cisplatin-induced ototoxicity.

 4) Please check and remedy the titles and labeling throughout methods and materials section and the results section as per PLOS One guidelines. Also, figure legends should be a separate section and not included within the results section.

The manuscript heading has been modified to match the guidelines. But in the case of figure legends, guideline says: “Figure captions are inserted immediately after the first paragraph in which the figure is cited. Figure files are uploaded separately.” So we left the figure legend as it is. 

5) Results: Explain better the significance of testing NRF2 nuclear to cytoplasmic ratio.

The significance of nuclear translocation of NRF2 has been added to the 'Nuclear translocation of NRF2 by LA' section of the results.

Added sentences

Normally NRF2 is negatively regulated by a cytosolic protein called KEAP1 (Kelch-like ECH associated protein 1). However, when activated by oxidative stress, NRF2 is released from KEAP1 and migrates from the cytoplasm to the nucleus to promote the expression of antioxidant proteins (25). We observed whether the migration of NRF2 to the nucleus increased in HEI-OC1 cells when treated with LA, and we also examined whether inhibition of NRF2 eliminates the antioxidant activity of LA.

6) Figure 2B: Please provide a key for the graph curves depicted.

Figure 2B shows the result of the bar graph of figure 2A at 24 hours. But the contents of figure 2B overlaps with figure 2A and the difference between the experimental groups is not visually significant. So we removed it.

 7) All figures should be properly labeled.

 8) Figures are poor quality. Please check guidelines for figure submission.

The responses to comment 7 and 8. We have modified figure 3, 5, 6, and 7 at the request of another reviewer. During the modification, we also reviewed the label and quality of the figure again. Thank you! 

 9) Result 2-1: Nuclear translocation of NRF2 by LA: What is the figure pertaining to this? If this is the immunofluorescence image on the penultimate page, this is not properly explained nor labeled. What is the graph with curves depicting? Where is it described and where is the key?

Figure 5 is the figure associated with Results 2-1. Maybe there was a misunderstanding. The immunofluorescence picture you mentioned (Figure 6) is related to Result 2-2: Reduction of the antioxidant and anti-inflammatory actions of LA by NRF2 inhibition. 

 10) Results: Figure 5A and 5B: Please include a LA only control and a siNRF2 only control in both these experiments.

Thank you for your comment. We added the experimental group you mentioned and modified result.

---

## [Decision Letter · Decision Letter 1]

6 Dec 2019

α-Lipoic acid prevents against cisplatin cytotoxicity via activation of the NRF2/HO-1 antioxidant pathway

PONE-D-19-21584R1

Dear Dr. Kim,

We are pleased to inform you that your manuscript has been judged scientifically suitable for publication and will be formally accepted for publication once it complies with all outstanding technical requirements.

With kind regards,

Partha Mukhopadhyay, Ph.D.

Section Editor

PLOS ONE

Additional Editor Comments (optional):

Reviewers' comments:

Reviewer's Responses to Questions

**Comments to the Author**

1. If the authors have adequately addressed your comments raised in a previous round of review and you feel that this manuscript is now acceptable for publication, you may indicate that here to bypass the “Comments to the Author” section, enter your conflict of interest statement in the “Confidential to Editor” section, and submit your "Accept" recommendation.

Reviewer #1: All comments have been addressed

Reviewer #2: All comments have been addressed

2. Is the manuscript technically sound, and do the data support the conclusions?

Reviewer #1: Yes

Reviewer #2: Yes

3. Has the statistical analysis been performed appropriately and rigorously? 

Reviewer #1: Yes

Reviewer #2: Yes

4. Have the authors made all data underlying the findings in their manuscript fully available?

Reviewer #1: Yes

Reviewer #2: Yes

5. Is the manuscript presented in an intelligible fashion and written in standard English?

Reviewer #1: Yes

Reviewer #2: Yes

6. Review Comments to the Author

Reviewer #1: The authors have made significant effort to address previous concerns and I have no further questions.

Reviewer #2: (No Response)

7. PLOS authors have the option to publish the peer review history of their article (what does this mean?). If published, this will include your full peer review and any attached files.

Reviewer #1: No

Reviewer #2: No

---

## [Editor Report · Acceptance letter]

10 Dec 2019

PONE-D-19-21584R1 

α-Lipoic acid prevents against cisplatin cytotoxicity via activation of the NRF2/HO-1 antioxidant pathway 

Dear Dr. Kim:

I am pleased to inform you that your manuscript has been deemed suitable for publication in PLOS ONE. Congratulations! Your manuscript is now with our production department. 

With kind regards,

on behalf of

Dr. Partha Mukhopadhyay 

Section Editor

PLOS ONE